# Purification and Characterization of the Enzyme Fucoidanase from *Cobetia amphilecti* Utilizing Fucoidan from *Undaria pinnatifida*

**DOI:** 10.3390/foods12071555

**Published:** 2023-04-06

**Authors:** Shu Liu, Qiukuan Wang, Zhenwen Shao, Qi Liu, Yunhai He, Dandan Ren, Hong Yang, Xiang Li

**Affiliations:** 1Colleage of Food Science and Technology, Huazhong Agriculture University, Wuhan 430070, China; 2Key Laboratory of Aquatic Products Processing and Utilization of Liaoning Province, National R and D Branch Center for Seaweed Processing, College of Food Science and Engineering, Dalian Ocean University, Dalian 116023, China; 3Qingdao Seawit Life Science Co., Ltd., Qingdao 370200, China; 4Bureau of Science and Technology of Qingdao West Area, Qingdao 266555, China

**Keywords:** enzymatic activity, enzymatic properties, fucoidanase, low molecular weight, ultrasonication

## Abstract

Fucoidanase is an unstable enzyme with high specificity that requires a large about of time to screen it from microorganisms. In this study, enzymatic hydrolysis was used to produce low-molecular-weight fucoidan from microorganisms via the degradation of high-molecular-weight fucoidan without damage to the sulfate esterification structure of oligosaccharide. The microbial strain HN-25 was isolated from sea mud and was made to undergo mutagenicity under ultraviolet light. Fucoidanase was extracted via ultrasonication and its enzymatic activity was improved via optimization of the ultrasonic conditions. The enzymatic properties and degradation efficiency of fucoidanase were characterized. The microbial strain HN-25 is a Gram-negative aerobic and rod-shaped-cell bacterium, and therefore was identified as *Cobetia amphilecti* via 16s rDNA. The results proved that fucoidanase is a hydrolytic enzyme with a molecular weight of 35 kDa and with high activity and stability at 30 °C and pH 8.0. The activity of fucoidanase was significantly enhanced by sodium and calcium ions and inhibited by a copper ion and ethylenediaminetetraacetate (EDTA). There was a significant decrease in the molecular weight of fucoidan after enzymatic hydrolysis. The low-molecular-weight fuicodan was divided into four fractions, mainly concentrated at F3 (20~10 kDa) and F4 (≤6 kDa). These consequences suggest that fucoidanase obtained from *Cobetia amphilecti* is stable and efficient and could be a good tool in the production of bioactive compounds.

## 1. Introduction

Fucoidan is a class of well-known natural sulfated polysaccharide from brown algae and sea invertebrate animal echinoderms that exhibits a variety of pharmacological activities such as antioxidation, immunomodulation [1], hypolipidemic activities [2,3], anticoagulant [4], antiviral [5] and hepatoprotective effects [6] and the combatting of carcinogens. Fucoidan is a structurally diverse group of sulfate with high molecular weight (HMW) and a unique chemical structure. The polysaccharide backbones in fucoidans are distinguished to type I or type II. Type I chains consist of reiterating (1→3)-linked α-L-fucopyranose residues, whereas type II chains comprise the interchanging (1→3) and (1→4)-linked α-L-fucopyranose residues [3,7,8]. Fucoidan is also composed of monosaccharides such as fructose, mannose, galactose, xylose, rhamnose and uronic acids. Due to the complex architectures of algal fucoidan [9], the structure–activity relationship of fucoidan remains unknown. With its biological activities, the application of fucoidan is still limited due to its HMW, complex chemical composition and low solubility and absorption. Fucoidan with HMW is very difficult to be absorbed and utilized by the human body compared with low-molecular-weight fucoidan (LMWF). Therefore, the use of HMW fucoidan in clinical drugs is limited. Meanwhile, LMWF is relatively homogeneous, easy to absorb and has strong functional activity and antigenicity.

It is known that biological activity is often manifested by oligomeric fragments of polysaccharides [10,11]. Several studies have suggested that LMWF exhibits biological activities such as anti-inflammatory [12], antiangiogenic [13], a hypoglycemic effect [14] and antioxidant activity [15,16]. In addition, LMWF has also been shown to have a protective effect against hypertension, hyperlipidemia and the hyper-responsiveness of aortic smooth muscle in type I diabetic rats [17]. LMWF has great potential as an adjuvant treatment for cardiovascular complications in type I diabetes mellitus [18]. Moreover, a previous study demonstrated that LMWF is a candidate drug against diabetic peripheral arterial disease (PAD) [19]. The biological activities of LMWF are linked to its specific chemical structure such as its molecular weight [20], composition (i.e., the degree and location link of sulfation and monosaccharide) [21] and structure (i.e., backbone structure, the degree of branching and substitution, chain conformation, etc.) [22,23].

Fucoidan with a high degree of sulfation and LMW exhibits stronger anti-diabetic activity, while fucoidan derivatives with persulfuric acid display strong hypoglycemic activity [24]. Furthermore, studies have shown that only fucoidans with a 1,3 and 1,4 carbon chain have a greater inhibitory effect on α-D-glucosidase [24].

Fucoidanases are enzymes that catalyze the hydrolysis or degradation of fucoidans. Fucoidanases can hydrolyze fucoidan to produce sulfated LMWF without the removal of its side substitute groups. Fucoidanases are either exo- or endo-acting enzymes. Endo-fucoidanases would be the optimum choice for the preparation of oligosaccharides of fucoidan. They could additionally be used as a preferred tool to elucidate the structure of fucoidan [25,26] and avoid the hydrolysis of the sulfate groups of fucoidans to minimize any adverse effects on the activity of polysaccharide. However, the commercial availability of fucoidanase is scarce due to a relatively trivial study on the enzyme. Meanwhile, existing studies have shown that the activity of fucoidanase is much lower than that of other similar polysaccharide enzymes, including alginate lyase.

Many studies have indicated that fucoidan could be degraded by some marine bacteria [27,28], and the hydrolases responsible for the degradation have been observed in the enteric and digestive glands of marine animals [29]. Fucoidanase comes from a wide range of sources, but the most important ones of fucoidanase are marine enzyme-producing microorganisms which are usually screened from algae, marine mud, seawater and the viscera of marine animals [30,31]. Moreover, fucoidanase has been shown to exist in marine molluscs such as *Lambis* sp. [32], *Littorina kurila* [33], marine echinoderms, the pancreas of *Patiria pectinifera* [34], sea cucumber, marine urchins such as *Strongylocentrotus nudus* [35], marine shellfish such as *Haliotus* sp. [36], *Patinopecten (Mizuhopecten) yessoensis* [37] and marine bacteria such as *Fucobacter marina* SI-0098 (*Flavobacterium* sp. SA-0082) [38], *Flavobacterium algicola* [17] and *Vibrio* sp. Other groups of marine organisms in which fucoidanase occurs are N-5 [39], marine proteobacteria such as *Pseudoaltero monascitrea* KMM 3296, KMM 3297 and KMM 3298 [28], *Alteromonas* sp. SN-1009 [25], *Luteolibacter algae* (H18) [40], *Formosa haliotis* [41], marine fungi such as *Dendryphiella arenaria* TM94 [26], the *Mucor* sp. 3P fungal strain [42] and six algal-inhabiting fungi used in the fermentation process [43], etc.

The microorganisms were selected by judging the degradation of fucoidan, monitoring microbial growth on specific carbon source medium. However, the fucoidanases produced by these microorganisms only exhibit low activity. For example, the crude fucoidanase production by *Dendryphiella arenaria* was 3.43 U/mL and 4 U/mL [44]. Thus, an optimization study to improve fucoidanase production and characterization in marine microorganisms is essential. Screening microorganisms capable of producing highly active enzymes such as fucoidanase for the extraction and purification of this enzyme is of great significance. Meanwhile, information on the enzymological properties and degradation products of fucoidanase is lacking. Therefore, in this study, microorganisms producing fucoidanase were screened from seawater, marine mud and the viscera of marine animals, utilizing fucoidan from *Undaria pinnatifida* as a carbon source in order to clarify the activity of the enzyme, its properties and characteristics under specific ultrasonic conditions. In addition, the separation and purification of the degraded polysaccharides were carried out in this paper, which laid a foundation for the analysis of the structure–activity relationship of degraded polysaccharides and provided a theoretical foundation for the application of fucoidanase.

## 2. Materials and Methods

### 2.1. Extraction of Fucoidans

Fucoidan was extracted from *Undaria pinnatifida* as described by a previous study [45] through enzymatic hydrolysis and ethanol precipitation. The filtered seawater and samples of marine organisms such as sea cucumber, sea urchin, sea mud, sea sand, sea water and starfish were collected from different sea areas of Dalian, China.

### 2.2. Screening and Identification of Strains of Marine Organisms

The samples of microorganisms collected from echinoderm viscera, sea mud, sea sand and seawater, respectively, were diluted using saline water under aseptic conditions. Thereafter, 100 μL of diluted liquid dissolved in filtered seawater was added to the solution, spread on screening plates (0.2% of fucoidan, 0.2% of ammonium, 0.2% of nitrate and 0.2% of agar, dissolved using filtered seawater and then sterilized) and incubated. After incubation at 25 °C for 72 h, we selected strains with good growth states and different morphologies via plate streaking. The obtained strains from the initial screening were rescreened via inoculation in 50 mL of rescreening liquid medium (0.2% of fucoidan, 5 mg/mL of ammonium nitrate, 0.2% of peptone, dissolved in filtered seawater and then sterilized). The media were incubated at 150 rpm and 25 °C for 72 h in a constant temperature oscillator. The content of fucoidan was determined using the Methylene Blue method of Soedjak H S [46] with slight modifications. Fucoidan solutions were thereafter diluted to 0.02% via an assay procedure using 1 mL of reaction solution. The fermented broth (1 mL) of each strain was centrifuged to obtain 50 μL supernatant, and then the absorbance was measured at 559 nm using a spectrophotometer with the blank medium serving as a control. The degradation rate of fucoidan was determined from a standard curve. The strains with a degradation rate over 10% were selected as the target strains and were stored in 20–30% glycerol medium at −80 °C for further analysis. The species identity of the strains was identified using 16s rDNA sequencing [47]. DNA was isolated from the bacterial culture and its quality was evaluated in 1% agarose gel. The isolated DNA was amplified using 16s rDNA specific primers (7F&1492R). Polymerase chain reaction (PCR) was carried out using a standard procedure [48] with some modifications. Products obtained from the PCR were sequenced at Sangon Biotech (Shanghai, China) Co., Ltd. The DNA sequences were submitted to the GenBank database. The homology of sequences with 12 strains of the genus Halomonas and one strain of the genus Cobetia was analyzed using the NCBI BLAST retrieval system. Phylogenetic trees were constructed using the neighbor joining method with MEGA software version 6.0 (Mega Limited, Auckland, New Zealand).

### 2.3. Mutagenesis of Strains of Marine Organisms

Mutagenesis of strains of marine organisms was carried out using ultraviolet (UV) light. The bacterial suspensions obtained from *Cobetia amphilecti* were cultured for 24 h and centrifuged at 6000 rpm for 10 min. The supernatant was decanted, and the residue was diluted to 106 cells/mL in 0.9% saline solution (*w*/*v*). Mutagenesis was conducted by placing 15 mL of the diluted solution in a 90 mm Petri dish under UV lamp (40 W) with a vertical 30 cm irradiation distance and exposure of 0–180 s irradiation time. Then, cell suspension (0.1 mL) was spread in solid plate medium and left in the dark for 72 h at 25 °C. The strains with a mortality rate of 70–90% were selected using the plate counting method and the fatality curve was drawn. The residual colonies were picked and transferred to 10 mL of fermented medium, and cultured at 25 °C with 150 rpm for 72 h. The enzymatic activity was measured with the original strain as the control. Mutants with higher enzymatic activity and good genetic stability were selected as the target strains.

### 2.4. Enzyme Production and Growth Monitoring

The bacteria strain stored at −80 °C was used for enzyme production. This was allowed to be thawed at room temperature, and then inoculated in recovery solid medium (0.2% fucoidan of *Undaria pinnatifida*, 0.2% peptone and 2% agar, dissolved in filtered seawater) at 25 °C for 72 h. The single colony on the plate was transferred to 15 mL liquid medium (0.2% fucoidan of *Undaria pinnatifida* and 0.2% peptone dissolved in filtered seawater) and incubated at 25 °C for 24 h in a shaking water bath with 150 rpm for the seed culture. Then, the strain was inoculated in 50 mL of 5% (*v*/*v*) of the liquid medium. The enzyme was thereafter produced at 25 °C and 150 rpm for 72 h. The biomass (OD 600) and enzymatic activity were measured every 12 h for a period of 120 h. The growth of strains and enzyme production curves were extrapolated.

### 2.5. Enzyme Extraction

The extraction of enzymes was performed via ultrasonication and was improved via optimization of the ultrasonic conditions. The post-culture fluid was centrifuged with 10,000 rpm at 4 °C for 10 min. The supernatant (extracellular enzyme) and precipitate (bacterial cells) were collected, respectively. The precipitated bacterial cells were redissolved in a C-buffer solution (pH 8.0, 20 mmol/L Tris-HCl including 0.2 mol/L NaCl) to control the concentration of the bacterial solution. The ultrasonication treatment was carried out with a UV lamp of about 300 W for 15 min (single time 5 s, interval time 5 s). The mixture was thereafter centrifuged at 10,000 rpm for 10 min. The supernatant was collected as crude fucoidanase (intracellular enzyme), and the precipitation was redissolved for further monitoring.

Experiments with single-factor and orthogonal design were carried out for the optimization of ultrasonic parameters, including ultrasonic power, concentration of bacteria, total treatment time, ultrasonic single time and interval time factors in the ultrasonic process.

### 2.6. Preliminary Purification of Fucoidanase

Precooled acetone (−20 °C) at different volumes was added into 20 mL of crude enzyme solution, stirred slowly, placed at −20 °C for 1 h and then centrifuged at 8000 rpm and 4 °C for 10 min. The supernatant was decanted, and the precipitate was redissolved in 5 mL of T-buffer (pH 8.0, 20 mmol/L Tris-HCl) to determine enzymatic activity and protein content. The reaction mixture was dialyzed against distilled water for 48 h (MWCO 1000 Da). The residue was lyophilized to obtain the fucoidanase.

### 2.7. Assay of Enzymatic Activity

The activity of fucoidanase was monitored using the method of Ying Wang [49] with minor modifications. One unit (U) of fucoidanase activity was defined as the amount of enzyme that liberated 1 μmol of L-fucose from 1 mL of enzyme mixture per min. The reaction mixture consisted of 0.2% fucoidan solution dissolved in T-buffer and fucoidanase solution in an equal volume. After incubation with 120 rpm in a constant temperature oscillator at 30 °C for 10 min, the enzyme was deactivated in boiling water for 20 min, and then the content of reducing sugar was measured according to the potassium ferricyanide method [49], which is a colorimetric assay determining the content of reducing sugar released from fucoidan after incubation with fucoidanase [31]. The enzyme samples deactivated in boiling water for 20 min before incubation and centrifugation served as the control. The enzymatic activity was determined using the following formula:(1)$$Y(U/\mathrm{mL})=\frac{4\times (x_{1}-x_{0})}{10}$$
where *Y* (U/mL) is the enzyme activity per milliliter of enzyme solution, *x*_1_ is the fucose content of the test samples, *x*_0_ is the fucose content of the control, 4 is 1 divided by 0.25 mL of sample volume and 10 is the time of 10 min.

The standard curve of fucose is *An* = −0.2348 *Xn* + 0.5611, where *Xn* (mg/mL) is the fucose content of the control and test samples and *An* is the optical absorbance of the control and test samples at 420 nm.

### 2.8. Determination of Enzymatic Characteristics

The fucoidanase was characterized by determining the activity and stability of the enzyme under various temperatures (20–50 °C) and pH values (5–10). The promoting or inhibiting factors of enzymatic activity were determined by the effect of metal ions (Na^+^, Mg^2+^, K^+^, Ca^2+^ and Cu^2+^) and EDTA on the reaction mixture. Purified enzyme samples (dissolved in solutions containing different metal ions) were mixed with 0.2% fucoidan, and the final concentration of metal ions was 5 mmol/L. Then, the enzyme activity was measured separately via the potassium ferricyanide method, as described in Section 2.7. The highest activity of the enzyme was regarded as 100% for the determination of the relative enzymatic activity.

The SDS-PAGE (SDS-polyacrylamide gel electrophoresis) was determined to identify the purity and estimate the molecular weight of fucoidanase as described by the method in Chen’s work [50].

### 2.9. Preliminary Determination of Mode of Fucoidanase Degradation

The mode of fucoidanase degradation was (either lyase or hydrolase) preliminarily determined by monitoring the change in unsaturated chemical bonds during the enzymatic reaction.

Fucoidan with 0.2% concentration (prepared with T-buffer) was mixed with purified enzyme 2:1 (*V*:*V*). The final NaCl concentration obtained in the reaction system was adjusted to 0.1 mol/L. The reaction mixture was incubated at 30 °C in a shaking water bath with 120 rpm for 24 h. Samples were taken at 0, 1, 2, 4, 6, 8, 10, 12 and 24 h, and the absorbance was measured directly at 232 nm. The enzyme that was deactivated at 80 °C for 20 min served as the control.

### 2.10. Purification of Fucoidanase

The purification of fucoidanase was carried out using column chromatography. Enzyme solution (5 mL) obtained via acetone precipitation was loaded on a DEAE-Sepharose Fast Flow column (2.6 × 30 cm) at 4 °C and the mobile phase parameters were the following: 0–4 h, Tris-HCl buffer (pH 8.0); 4–21 h, NaCl (0–1.3 M). The flow rate was 0.5 mL/min, and an automatic fraction collector was used to collect the elution (3 mL/tube). The enzyme activity and protein content of the collected fractions were measured. The elution curve was determined by plotting the tube number against the enzymatic activity and protein content.

### 2.11. Preparation of Enzymatic Solution of Fucoidan

The enzyme solution and the fucoidan solution (2.0%) were mixed at 1:1 and placed in a water bath at 30 °C for 24 h. Then, they were inactivated in a boiling water bath and centrifuged (10,000 r/min, 10 min, at 4 °C) to remove the precipitation and obtain enzymatic solution of fucoidan.

## 3. Results and Discussion

### 3.1. Sequence Analysis of 16s rDNA

A total of 82 single colonies with different morphologies were obtained in the preliminary screening of bacterial strains from different sources. Fucoidan was the only carbon source in the rescreening medium, while the change in content in the post-culture fluid indicates the degradation ability of fucoidan of the strain. Only three strains had a degradation rate of more than 5%. The three strains were HN-25 with a degradation rate of 10%, HSE-1-1 with a degradation rate of 12.2% and HX-3 with a degradation rate of 7.8%, respectively. These three strains therefore were taken as the target strains. Molecular identification of the bacteria strain on the basis of the 16s-rDNA-based molecular technique [47] showed that HN-25, HSE-1-1 and HX-3 amplified via PCR had bands of 1441, 1456 and 1448 bp, respectively. Therefore, HSE-1-1 was identified as *Vibrio splendidus*, while HX-3 and HSE-1-3 were identified as Shewanella, both of which are putrefying bacteria. Moreover, in this study, these two microorganisms were not regarded as functional strains. The strain HN-25 with the degradation rate of 10% was selected as the target strain. Meanwhile, 13 strains, including *Cobetia amphilecti* (KMM 296), *H. aidingensis* (GQ281062), *H. alkaliantarctica* (AJ564880), *H. alimentaria* (AF211860), *H. Almeriensis* (AY858696) and *H. andesensis* (EF622233) were obtained and selected using the BLAST retrieval system (Figure 1).

The homology of HN-25 to the sequence of *Cobetia amphilecti* was 100% via 16s rDNA. HN-25 was identified as *Cobetia amphilecti* and the morphological and biochemical characteristics of the strain HN-25 were Gram-negative, aerobic, non-pigmented and rod-shaped-cell microorganisms. HN-25, therefore, was identified as *Cobetia amphilecti*, which is a novel species belonging to the genus Cobetia [51]. Studies have shown that glutaminase-free L-asparaginase [52], alkaline phosphatase/phosphodiesterase [53] and poly (3-hydroxybutyrate) (PHB) [54] were found in *Cobetia amphilecti*. The nucleolytic enzymes obtained from *Cobetia amphilecti* KMM 296 have also shown antibacterial activity [53]. There have been no previous reports obtained on the fucoidanase activity from *Cobetia amphilecti*. From the results, this is the first time that *Cobetia amphilecti* has been confirmed as a fucoidanase-producing strain.

### 3.2. Production and Selection of Fucoidanase

In order to improve the efficiency of fucoidanase degradation activity, HN-25 was first mutated via UV irradiation. After mutagenesis, the activity of fucoidanase in HN-25-M was increased by 67.4%. This is in accordance with the results obtained by Wang et al. [55] who reported that the fucoidanase activity of strain Rc2-3-Mut obtained via the UV irradiation of Flavobacteriaceae RC2-3 increased by 40.5%. The results showed that mutagenesis via UV was an effective method to improve the activity of fucoidanase.

The growth curves of strain and enzymatic activity were monitored during the cultivation process (Figure 2). The crude enzyme liquid was prepared after centrifugation, ultrasonication, etc. Enzyme activities in process products, such as cultivation broth, exoenzyme, endoenzyme and residue, were determined, respectively, and meanwhile the target enzymes were collected according to the enzyme activities. The enzymatic activity increased gradually as the microbial growth attained the logarithmic phase and reached its maximum when the bacterial biomass was ≤1, followed by a stable state of microbial growth. The results obtained from the growth curve showed that the best time to extract the enzyme was when the microbial growth attained the middle of the logarithmic stage. The results also showed that the endoenzymatic activity was higher than the exoenzyme and residue and equivalent to the enzymatic activity in the cultivation broth, while the exoenzymatic activity was lower, and no enzymatic activity was obtained in the residue sediment. The results showed that the enzyme activities in cultivation broth mainly came from the endoenzyme. Considering the preparation and purification, the exoenzyme was mixed with medium, making it difficult to separate. Therefore, endoenzymes were selected as the target enzymes for the study. According to the processes shown in Figure 2, the strain can be screened and fucoidanase can be extracted and prepared. In addition, the enzymatic activity is the highest in the logarithmic phase, and the enzyme that degraded fucoidan was endoenzyme, which has high activity and a feasible preparation process.

### 3.3. Optimum Conditions and Factors Affecting Enzymatic Activity during Ultrasonication

The effects of ultrasonic power, bacterial concentration, single ultrasonic time and interval time on enzymatic activity are shown in Figure 3. There was an increase firstly and then a decrease in the enzymatic activity as ultrasonic power increased. When the ultrasonic power was about 300 W, the enzymatic activity reached its maximum value (Figure 3A). This may be a result of the incomplete crushing effect which led to an incomplete release of intracellular enzymes in the cell when the crushing power was low. Increasing the power was conducive to the formation of cavitation bubbles, thus enhancing the ultrasonic effect [56]. When the crushing power was high, cavitation strength became saturated, which resulted in bubbles that were not useful. Consequently, the scattering attenuation increased and the cavitation intensity decreased, which further led to a decrease in enzymatic activity. Meanwhile, the total ultrasonic time had a significant effect on the activity of fucoidanase (Figure 3B). The ultrasonic time was too short to generate relatively high activity of the enzyme, meaning that some cells were not completely broken. With an increase in the total time, the enzyme amount increased and the enzymatic activity was significantly improved at 9–15 min. However, the total crushing time was too long (18 min), which led to the enhancement of the ultrasonic thermal effect on the enzymatic activity, leading to the deactivation of the enzyme. Considering the above reasons and time cost, the total time was fixed at 9 min. As the concentration of bacterial suspension became low, the amount of enzyme released via the ultrasonic treatment decreased (Figure 3C). With an increase in the concentration of bacterial suspension, the amount of enzyme increased and the enzymatic activity increased gradually. The enzymatic activity reached its maximum when the concentration of bacterial suspension was 30 mg/mL. It was difficult for the enzymatic activity to increase when the concentration was continuously increased. This was because the concentration of bacteria was too high, energy was damaged in the transfer process of the ultrasonic wave and the effect of cell breakage was poor.

The extraction effect was high and the single time was set at 3 s (Figure 3D). Meanwhile, a short single time might lead to lower extraction and weak enzymatic activity. However, if the single time was too long, it could lead to the deactivation of some enzymes due to the fact that the ultrasonic process generated a lot of heat in a short time that was difficult to cool down easily. Due to the thermal effect produced via the ultrasonic process, bacterial suspension should be placed in an ice bath during treatment. This implies that setting up an appropriate interval time could be an effective means for eliminating the thermal effect. The enzymatic activity presented a parabolic state within the interval of 1 to 5 s and reached its peak at 3 s (Figure 3E). This may be due to the increase in the interval time of the ultrasound which reduced the ultrasonic frequency at a certain period, leading to incomplete cell breakage.

Orthogonal experiments with four factors and three levels were set up based on single-factorial experiments with the interval time durations fixed at 4 s as Appendix A. From the results obtained obtained from Appendix A, the influence order of experimental factors on enzymatic activity was as follows: ultrasonic power > total time > concentration of bacteria > single time. After the verification experiment (Appendix A), the ultrasonic power for the optimal extraction state was 300 W, the concentration of bacteria was 40 mg/mL, the total time was 12 min and the single time was 5 s. The highest enzymatic activity was reached at 275 U under the optimal reaction conditions. In addition, the enzymatic activity was also related to the growth state of the cultured microorganisms. In conclusion, it can be seen from Figure 3 that in the process of enzyme extraction, ultrasonic conditions have a significant effect on the enzymatic activity, and the research results can provide guidance for the efficient extraction of fucoidanase.

### 3.4. Purification of Fucoidanase via Acetone Precipitation

Table 1 shows the activity of fucoidanase before and after purification, the specific activity and the yield of enzymatic activity. The results revealed that after purification with a volume ratio of enzyme solution to acetone of 1:2.0, the loss in total enzymatic activity was small, the protein was purified, the specific activity was increased by 1.21 times compared with the unpurified enzyme and the yield of enzymatic activity reached 98%. Therefore, to purify fucoidanase via acetone precipitation, the optimal volume ratio of enzyme liquid to acetone required was 1:2.0, and the specific activity was 0.43 U mg^−1^. Burtseva et al. [57,58] isolated enzymes with fucoidan hydrolase activity from 33 species of marine invertebrates, and obtained activities between 0.20 and 0.26 U mg^−1^ protein. Rodríguez-Jasso [42] obtained fucoidanase with *Mucor* sp. 3P grown on autohydrolyzed alga substrate under agitation, and the fucoidanase activity achieved was 0.37 U mg^−1^ protein after 72 h of cultivation. In general, the enzymatic activity obtained in this paper is consistent with that reported in the literature.

### 3.5. Characteristics of Fucoidanase

Figure 4 shows the crude enzyme precipitated via acetone, dissolved in buffer and purified via column chromatography using the DEAE-Sepharose Fast Flow column. The figure shows three peaks of the protein (P1–P3). The P2 and P3 protein peaks were the peaks of enzymatic activity after detection. These two protein peaks were the peaks of fucoidanases. The maximum peak (P2/E1) of the enzymatic activity was collected and used to purify fucoidanases. The results obtained in this study from SDS-PAGE after freeze-drying show that the molecular weight of fucoidanase was 35 kDa, the result was shown in Appendix A. However, fucoidanase might have different molecular weights in different organisms. For example, it was reported that the molecular weights of fucoidanases in E1, E2 and E3 of *Vibrio* sp. N-5 were 39, 68 and 68 kDa, respectively, via the SDS-PAGE [39]. Chen et al. [50] also reported that the molecular weight of fucoidanase was 41 kDa obtained from Flavobacteriaceae RC2-3, while Wang et al. [56] indicated that the molecular weight of fucoidanase produced via UV-mutagenized Flavobacteriaceae RC2-3 was around 91 kDa, which was different from the results obtained in previous studies. From the reports of other studies such as the fucoidanase from hepatopancreas of *P. yessoensis* [59], which was 100–200 kDa, the fucoidanase obtained from *Dendryphiella arenaria* TM94, which was about 180 kDa [26], *Luteolibacter algae* H18, which was 112 kDa [40] and the relative molecular mass of a purified fucoidanase (FNase S), which was estimated to be 130 kDa [60], it was observed that the molecular weight of fucoidanases determined in this study was lower than those documented in previous reports. This implies that the molecular weights of fucoidanase from different organisms and sources are likely to be different.

The effect of temperature on enzymatic activity is shown in Figure 5A. The suitable temperature for the activity of fucoidanase was 30 °C. The enzymatic activity decreased rapidly as temperature increased above 30 °C. When the temperature was higher than 40 °C, the enzyme was gradually deactivated. Therefore, the reaction temperature should be set at about 25–35 °C in order to ensure the efficiency of fucoidanase. The effect of pH on enzymatic activity is shown in Figure 5B. The enzyme showed higher activity at pH values of 7.0 and 9.0. These results were in line with the report that it was favorable for enzymolysis under neutral and weak alkaline conditions [61]. Three temperatures (25, 30 and 35 °C) were selected to explore the thermal stability of the enzyme within 24 h. The order of preservation in the stability of the enzyme was 30 °C > 35 °C > 25 °C (Figure 5C). Additionally, pH values of 7.0, 8.0 and 9.0 were selected to explore the pH-dependent enzyme stability within 12 h. The results indicated that the enzyme was relatively stable at pH 7.0 within a short reaction duration, although there was no significant difference in enzyme stability at pH 7.0 and 9.0 for longer reaction durations. At pH 8.0, the enzymatic activity was relatively stable at about 80% of the original enzyme (Figure 5D). This is in accordance with the results obtained by a previous study [61] on a C-terminal deletion mutant fucoidanase Fhf2Δ484, which was active at 20–45 °C and at pH 6–9 and had optimal activity at 37 °C and pH 8. Kim et al. [60] also reported that the optimum pH and temperature of FNase S, an endo-acting fucoidanase that degrades Miyeokgui fucoidan, were pH 6.0–7.0 and 40–45 °C, respectively.

The effects of different metal ions (Na^+^, Mg^2+^, K^+^, Ca^2+^, Mn^2+^ and Cu^2+^) and EDTA on the activity of fucoidanase are shown in Figure 5E. From the results obtained, Na^+^ and Ca^2+^ had activation effects on fucoidanase (Figure 5E). Na^+^ had the strongest activation effect and enhanced fucoidanase HN-25-M activity by 31.7%, because the enzyme originates from marine bacteria. Ca^2+^ enhanced the activity of the enzyme by 12.5%. Cu^2+^ and EDTA had significant inhibitory effects on fucoidanase. Meanwhile Cu^2+^ reduced the activity of HN-25-W by 92.1%. This variation in the effect of metallic ions on the activity of fucoidanase may be a result of the fact that the enzymatic properties of fucoidanase from different sources vary greatly, which is consistent with the results of taxonomic identification. This result is consistent with the results obtained on the effect of Cu^2+^ on the activity of tFda1B and endo-fuocidanases. Cu^2+^ completely deactivated tFda1B and decreased the activities of endo-fucoidanases from *Lambissp* [32] and *Formosa algae* KMM 3553T (FFA1 and FFA2) [62,63]. Meanwhile Cu^2+^ did not affect the endo-fucoidanase from *Haliotus* sp. [36]. Ca^2+^ promoted the activity of the fucoidanase from HN-25-W, increasing its activity by 12.5%. This is consistent with the report of previous reports showing that Ca^2+^ activated FFA1, FFA2 [62,63] and *Alteromonas* sp. SN-1009 [64]. Furthermore, Mg^2+^ caused an inhibitory effect by reducing the activity of fuocidanase by 8.6%. From previous reports, Mg^2+^ stimulates the activity of tFda1B, but it inhibits fuocidanase action on FFA1, FFA2 [62,63] and endo-fuocidanases from *Lambis* sp. [32] and shows no obvious impact on endo-fucoidanase in *Alteromonas* sp. SN-1009 [64]. K^+^ shows no obvious impact on HN-25-W, nor on endo-fucoidanase in *Alteromonas* sp. SN-1009 [64]. The effects of different metal ions on enzymatic activity vary with different enzymes. In addition, the activity of fucoidanase was the highest when the concentration of NaCl was 0.1 mol/L (Figure 5F).

### 3.6. Degradation Mode of Fucoidanase

Polysaccharide lyases are a class of enzymes that cleave specific glycosidic linkages, exist in acidic polysaccharides and result in depolymerization [65]. These enzymes act by eliminating the mechanism to produce unsaturated oligosaccharide with UV absorption at 232 nm. In this study, the absorbance 232 nm was thereby used to monitor the changes in unsaturated chemical bonds in the process of the enzyme reaction. Using this method, the type of enzyme digestion was preliminarily determined. The occurrence of unsaturated bonds in the process of the enzymatic reaction indicates that the enzyme is a lyase but not a hydrolytic enzyme [66].

The change in absorbance of the products in the enzymatic reaction at 232 nm within 24 h is shown in Figure 6. From the results obtained, it was observed that with the extension of the reaction time, the absorbance at 232 nm of the control and test samples decreased initially and then increased throughout the course of the experiment while both trends were very consistent. The enzyme which was first deactivated in boiling water for 20 min served as the control. From this study, it can be inferred that the increase in absorbance was not caused by the enzyme action product such as unsaturated oligosaccharide. Furthermore, the absorbance of the samples increased in the same range as that of the control, indicating that the products of the enzyme should not cause a change in the absorbance at 232 nm. It was therefore observed that the enzyme is not a lyase but a hydrolytic enzyme. Further studies are needed to elucidate the mode of action of this enzyme. There is a study that evaluated the production of fucoidan hydrolytic enzymes via two fungal strains (*Aspergillus niger* PSH and *Mucor* sp. *3P*) through solid state fermentation in a rotating drum bioreactor. Different algal biomasses (untreated, autohydrolyzed and microwave-processed seaweed *Fucus vesiculosus*) were used as the substrate. The result found that none of the fungal strains produced fucoidan-hydrolyzing enzymes from untreated alga [42]. FdlA and FdlB were two unclassified endo-fucoglucuronomannan lyases and were previously claimed to be lyases acting on manno-glucurono-linkages in fucoidan from *K. crassifolia* [67]. The degradation modes of enzymes vary with different microbial sources.

### 3.7. Molecular Weights of Degradation Products

LMWF was obtained via a series of enzymatic processes and was separated via Sephocryl-100 column chromatography. By comparing the original fucoidan with the degradation products, it was observed that fucoidan is distributed mainly in the molecular weight range of 70 kDa. The content of fucoidan in >70 kDa, 20–40 kDa and 10 kDa was very small, and the overall molecular weight was large. From the elution curve (Figure 7), the elution peak at 70 kDa of the degradation products decreased significantly, and elution peaks were also observed at 20–10 kDa and <6 kDa, indicating that most of the fucoidans from *Undaria pinnatifida* were degraded into LMWFs during enzymolysis. The results obtained from this study indicated that the fucoidanase from *Cobetia amphilecti* had high efficiency for the degradation of fucoidan. From the standard curve, the degraded polysaccharides were divided into F1 (70 kDa > M > 40 kDa), F2 (40 kDa > M > 20 kDa), F3 (20 kDa > M > 10 kDa) and F4 (M < 6 kDa) based on their molecular weights, while LMWFs were mainly concentrated in F3 and F4 fractions. This result is in line with that of Kim et al. [60] who reported that an endo-acting fucoidanase degraded Miyeokgui fucoidan into smaller-sized galactofuco-oligosaccharides (1–4 kDa). The oligosaccharides generated were resolved into seven distinct low-molecular-mass fractions via Bio-Gel P-4, with the relative molecular weights of 3312 Da (peak 1), 2494 Da (peak 2), 1699 Da (peak 3), 1543 Da (peak 4) and 1312 Da (peak 5) [60].

In addition, the sulfate content of crude fucoidan was 26.97%, and those of fragments of degraded polysaccharides varied from 22.73 to 37.38%. According to the results, some of the fragments had lower and the other of the fragments had higher sulfate contents than that of crude fucoidan. The study concluded that the difference in sulfate content might be caused by the separation and purification process. Therefore, it was speculated that the fucoidanase had little effect on the sulfate group and would not eliminate the ester group.

## 4. Conclusions

From this study, it was observed that the HN-25 is Gram-negative, aerobic, non-pigmented and rod-shaped-cell bacteria, and was identified as *Cobetia amphilecti* via 16s rDNA sequencing. The optimal extraction efficiency for fucoidanase in the ultrasonication process was obtained at 300 W ultrasonic power, 30 mg/mL of bacteria and single and interval time durations of 4 s, and the highest enzymatic activity reached 275 U. The ratio of reaction mixture after ultrasonication to acetone (*V*:*V*) was 1:2.0. It was observed that the specific activity of the enzyme was increased by 1.21 times compared with the unpurified enzyme, the yield of enzymatic activity reached 98% and little or no loss was observed. After purification via DEAE Sepharose Fast Flow column chromatography, the molecular weight of fucoidanase was 35 kDa. The optimal conditions for the stability of the enzyme in the reaction system were 30 °C and pH 8.0. In addition, Na^+^ and Ca^2+^ had activation effects on fucoidanase by promoting the enzymatic activity to 132% and 113%. The enzymatic activity was much better when the concentration of NaCl was 0.1 mol/L. Cu^2+^ and EDTA had significant inhibitory effects on fucoidanase. The enzymolysis showed that fucoidanase was a hydrolytic enzyme rather than a lyase. The overall molecular weight of fucoidan was reduced after enzymic degradation, and four fractions with different molecular weights were obtained via Sephacryl-100 gel chromatography. In conclusion, our results suggest that fucoidanase obtained from *Cobetia amphilecti* is stable and efficient and could be a worthy tool with feasible application in the production of bioactive compounds. Further study on the biological activities of LMWF components needs to be performed.

## Figures and Tables

**Figure 1 foods-12-01555-f001:**
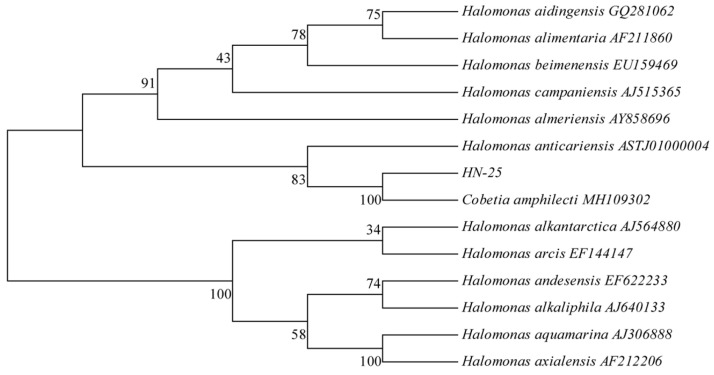
The 16s rDNA phylogenetic tree of strain HN-25.

**Figure 2 foods-12-01555-f002:**
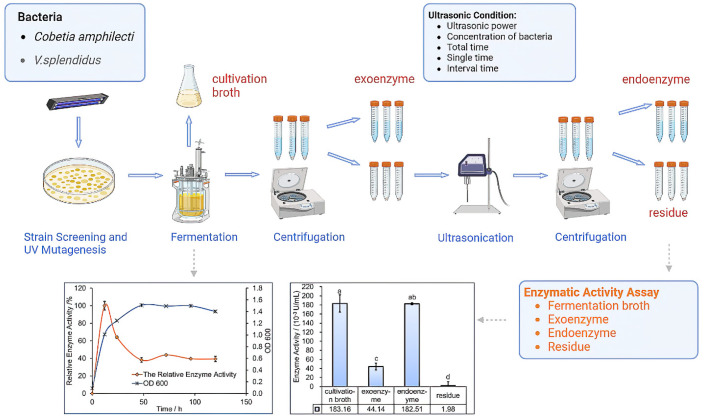
Strain screening and extraction of fucoidanase. The same letters of a, b, c and d mean the difference among groups is not significant, and different letters represent significant differences (*p* < 0.05).

**Figure 3 foods-12-01555-f003:**
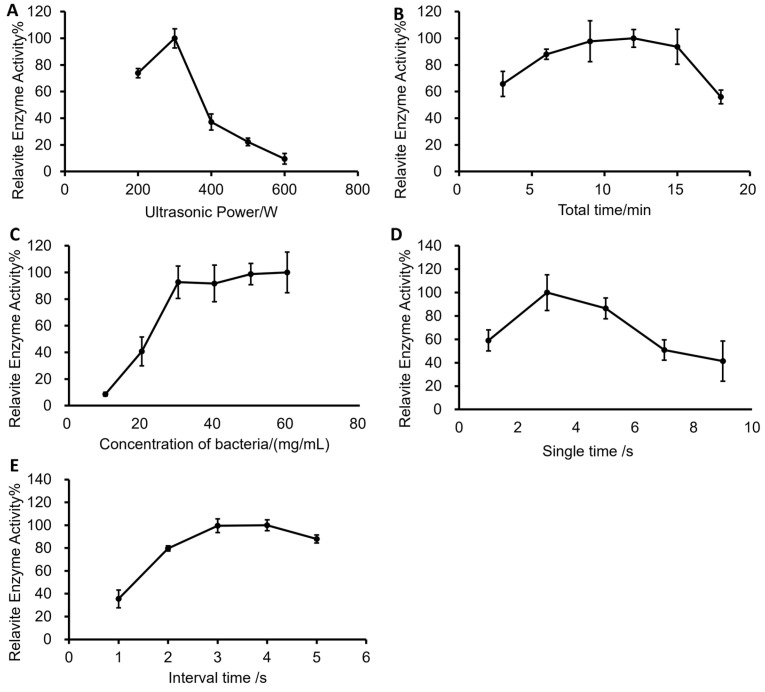
Optimum conditions and factors affecting enzymatic activity during ultrasonication. (**A**) Ultrasonic power. (**B**) Total time. (**C**) Concentration of bacteria. (**D**) Single time. (**E**) Interval time.

**Figure 4 foods-12-01555-f004:**
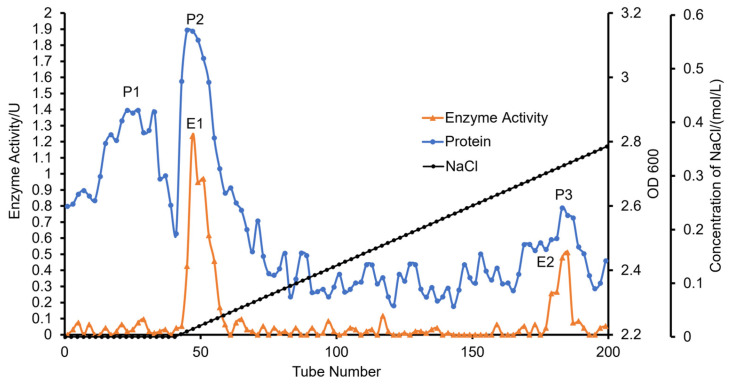
Elution curve of fucoidanase via DEAE Sepharose fast flow column chromatography. P1, P2 and P3 are three peaks of protein, and E1, E2 are two peaks of enzymatic activity.

**Figure 5 foods-12-01555-f005:**
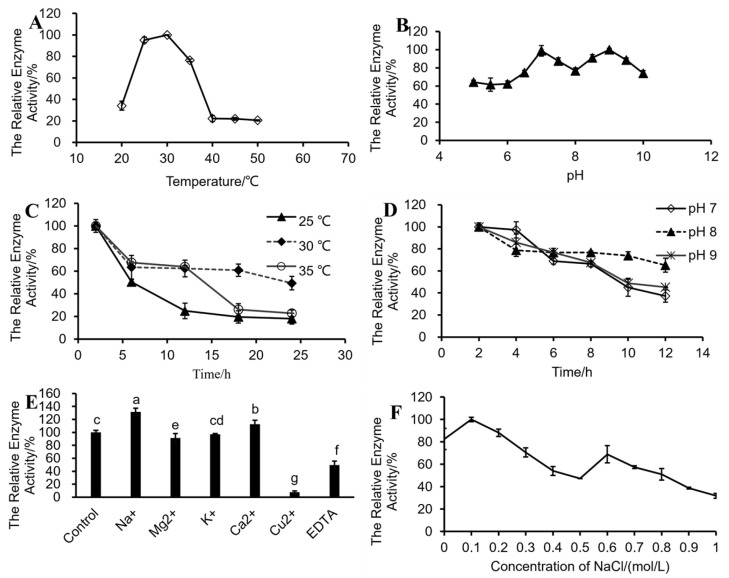
Effects of temperature, pH and metal ions on fucoidanase HN-25-M activity. (**A**) Effects of temperature on enzymatic activity. (**B**) Effects of pH values on enzymatic activity. (**C**) Effects of different temperature on stability on enzymatic activity. (**D**) Effects of different pH values on stability on enzymatic activity. (**E**) Effects of different metal ions and EDTA on enzymatic activity. (**F**) Effects of different concentrations of NaCl on enzymatic activity. The same letters of a~f mean the difference among groups is not significant, and different letters represent significant differences (*p* < 0.05).

**Figure 6 foods-12-01555-f006:**
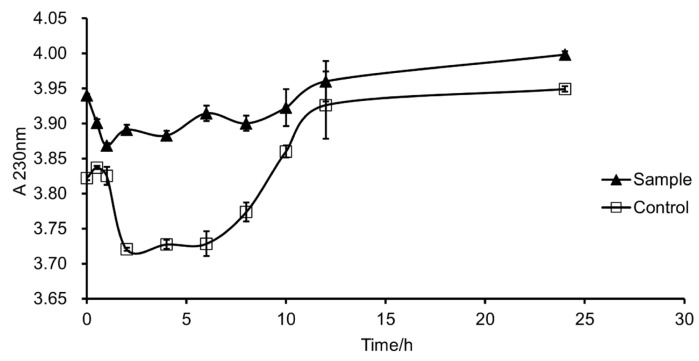
Degradation efficiency of fucoidanase at 232 nm.

**Figure 7 foods-12-01555-f007:**
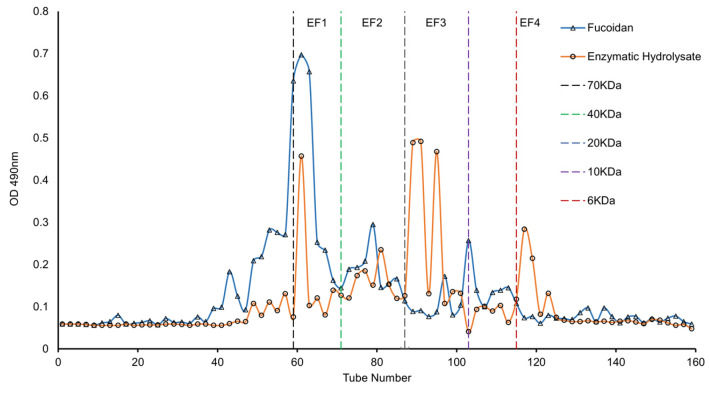
The elution curve of fucoidan and degradation products via Sephacryl-100 column chromatography.

**Table 1 foods-12-01555-t001:** Summary of purification of fucoidanase via acetone precipitation.

No.	Enzymolysis Liquid: Acetone (*V*:*V*)	Total Activity (U)	Specific Activity (U mg^−1^)	Purification (Fold)	Yield (100%)
1	1:0	6.60	0.41	1.00	100
2	1:1.0	2.49	0.16	0.39	38
3	1:1.5	3.48	0.25	0.60	53
4	1:2.0	6.44	0.43	1.21	98
5	1:2.5	4.44	0.38	0.93	67
6	1:3.0	5.56	0.38	0.92	84

## Data Availability

The data presented in this study are available upon request from the corresponding author.

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
