# Peer review of "Purification and Characterization of the Enzyme Fucoidanase from Cobetia amphilecti Utilizing Fucoidan from Undaria pinnatifida"

_foods, 2023, doi:10.3390/foods12071555_

Round 1

Reviewer 1 Report

The article is devoted to the preparation and characterization of the bacterial enzyme fucoidanase using fucoidan from Undaria pinnatifida as a substrate. The authors showed the effectiveness of this enzyme in the degradation of polysaccharides with the formation of lower molecular weight products more suitable for biomedical applications. There are several questions for revision:

1. Introduction, lines 32–35 and text: Fucoidans is the common name for sulfated fucans of brown algae, not echinoderms. These polysaccharides differ in structure and composition. Thus, the sentence should be modified like this: "Sulfated fucans are naturally occurring sulfated polysaccharides from brown algae (often referred to as fucoidans) and sea invertebrate animals Echinodermata...".

2. Section 2.7:  It is known that fucoidan from Undaria pinnatifida has a complex composition and structure. This is rather galactofucan having sulfate and acetyl ester groups. The assay of enzyme activity used is based on detection of liberated reducing sugars defined as L-fucose. However, this detection is not specific to fucose, it can be galactose or another sugar as well. A comparison with "classical" fucoidan from Fucus vesiculosus containing fucose only could help in this. Otherwise, releasing monosaccharides can be detected by more specific methods like TLC.

3. Section 3.7: Has the influence of the degree of sulfate/acetate esterification and the distribution of ester groups on the activity of fucoidanases been studied? Has it come to the elimination of ester groups under the action of these enzymes? These questions should be discussed on the basis of the data and/or literature obtained.

4. Section 3.8: The decrease in the molecular weight of the substrate is a good confirmation of the effective action of fucoidanases; however, it is also non-clear which part of the fucoidan is degraded - galactan or fucan itself.

Reviewer 2 Report

Overall comments:

1..  The manuscript is poorly written. The introduction paragraph must be improved in case of some unclear sentences and stylistic errors.

2. In the Introduction section the chemical insight into the structural features of fucoidan is missing. I suggest putting the scheme/figure with the types of polysaccharide backbones of fucoidans and marking possible sites of enzyme (fucoidanase) action.

3.   The purpose and significance of the research should be more emphasized according to the current literature reports.

4.     The whole manuscript contains many linguistic and grammatical errors; many informal and colloquial expressions were found in the text (to a large amount of these mistakes I did not mark them all in detailed comments). Extensive editing of the English language and style is required.

5.     Please improve the Figures in the whole text. Especially Figures 2 and 3 are too small and the descriptions are not readable.

6.     In the section Materials and Methods the descriptions of the enzymatic decomposition of fucoidan and the analytic method of determination of the molecular weights of obtained products are missing.

7.     I suggest to re-submit the paper once again after a detailed revision

8.     In my opinion, due to the fact that the main subject of the delivered manuscript is an enzyme and its characteristic I suggest transferring it to the other MDPI journal – Catalysts after a detailed revision.

Detailed remarks on the manuscript:

1.     Page 1, lines 13 – 14 – the sentence ‘Fucoidanase is unstable enzyme with high specificity and little damage to the structure of oligosaccharides that requires the high task to screen from microorganisms’ is unclear, please rewrite it. I would like to ask what the Authors meant by ‘little damage to the structure of oligosaccharides’.

2.     Page 1, line 16 – ‘HN-25’ – I would like to emphasize that if you use the abbreviation it should be explained previously in the text. I suggest putting the phrase ‘microbial strain’ before ‘HN-25’.

3.     Page 1, line 17 – change the word ‘enzyme’ to ‘enzymatic’ or ‘catalytic’.

4.     Page 1, line 20 – change the word ‘was’ to ‘is’.

5.     Page 1, line 36 – I would like to ask if you meant ‘sulfate’ or ‘sulphate’ instead of ‘sulpate’?

6.     Page 1, lines 41 – 42 – ‘Due to the complex architectures of algal fucoidan [9], the structural activity of fucoidan remains unknown’ – I would like to ask you to explain what you meant by ‘structural activity’ here?

7.     Page 2, line 79 – ‘sereened’ ??? – Did you mean here ‘secreted’?

8.     Page 3, Subsection 2.2, lines 110 – 114 – ‘The samples of microorganism collected from echinoderm viscera, sea mud, sea sand and seawater respectively were diluted with saline water under aseptic conditions. Thereafter, 100 μL of diluted liquid dissolved in filtered seawater was added to the solution and spread on screening plates containing 0.2% of fucoidan, 0.2% of ammonium, 0.2% of nitrate and 0.2% of agar, and then sterilized and incubated.’ – I would like to ask the Authors to read this description once again and check if they are sure that firstly they spread the microorganism sample on screening plates and then sterilized them. In my opinion, the microorganisms should be put on the previously sterilized plates to give them the possibility to grow.

9.     Page 3, line 118 – ‘…then sterilized’ – please follow the above remark no.8.

10.  Page 4, line 163; Page 5, line 227, and others – ‘the fermented solution’ – I suggest changing this phrase to ‘post-culture solution’ or ‘ post-culture fluid’.

Are you sure that the microorganisms’ cultivation can be named fermentation? Did you make this process under anaerobic conditions? From the manuscript can be read that it was an aerobic process. Please check the whole manuscript and change this description to the proper one.

11.  Page 4, line 178 – the time of centrifugation is missing

12.  Page 4, lines 165 and 179 – please explain what is the difference between the C-buffer and T-buffer.

13.  Page 4, lines 185 – 187 – the definition of enzyme catalytic activity is unclear. Please check it correctness.

Please remind that as standard, the international unit for the enzyme activity 1 U (μmol/min) is defined as the amount of the enzyme that catalyzes the conversion of one micromole of substrate per minute under the specified conditions of the assay method.

14.  Page 4, equation (1) – please explain the meaning of the used symbols (Y, x1, x0) and the numbers (5, 1000, 2)

15.  Page 5, Subsection 2.8, lines 197 – 203 – please change the phrase ‘… the stability of the enzyme…’ to ‘the activity and stability of the enzyme…’; give the temperature and pH range; give the information about what metal ions were tested. Describe in what conditions the enzyme activity was determined in the particular experiment.

16.  Page 5, line 237 – ‘ … strain HN-25 with the highest degradation rate was selected as the target strain’ – are you sure that this is true? According to the information from (Page 5, lines 229 –  230) the highest degradation rate possesses HSE-1-1 (12.2%), not HN-25 (10%).

17.  Page 6, line 256 – ‘In order to improve the efficiency of fucoidanase degradation, …’ – I suppose that you meant ‘degradation activity’ here instead of ‘degradation’.

18.  Page 6, lines 262 – 263 – ‘fermentation process’ – I suggest changing this phrase to ‘cultivation process’ according to the information given in the manuscript that C. amphiletcti is an aerobic bacteria.

19.  Page 6, lines 264, 272 and 274 – ‘fermented broth’ – this description should be also changed according to the above remark no.18.

20.  Page 7, lines 294 – 295 – ‘Furthermore, total ultrasonic time had no significant effect on the activity of fucoidanase (Figure 3b)’ – In my opinion, this statement is not in agreement with the results presented in Figure 3b. As can be seen, the total ultrasonic time has a significant effect on enzyme activity. Firstly it can be observed the activity increase from 100 to 140 U (time range 7 – 15 min) and after that, the activity drops to 80 U (18 min). Please analyze the results once again and improve the description.

21.  Page 7, lines 318 – 320 – ‘Orthogonal experiments with four factors and three levels were set up based on single factorial experiments. From the results obtained, the influence order of experimental factors on enzyme activity was as follows: …’ – I would like to ask the Authors to provide the methodology and the results obtained in this experiment to main text or supplementary materials.

22.  Page 8, lines 339, 341, 343 – Authors give the comparison of obtained activity results with the outcomes of previous research reports from the literature. Please check if the activity units were defined in the same way in all cited papers. If not there is not possible to directly state that the enzyme activity obtained in the presented manuscript is high without additional recalculation of these values.

23.  Page 8, line 348, Figure 4 – please mark the peaks of protein on the elution curve (Fig.4).

24.  Page 8, line 352 – provide SDS-PAGE results to main text or supplementary materials.

25.  Page 9, lines 364 – 366 – ‘it was observed that the molecular weight of fucoidanases determined in this study was higher than those documented in previous reports.’ – Please analyze if this statement is correct. In my opinion, 35 kDa obtained for the enzyme in this study is the lowest value in comparison to the cited literature papers,

26.  Page 9, lines 371 – 372 – ‘Meanwhile, the enzyme activity was inhibited when the temperature was lower.’ – I suggest Authors analyze the correctness of this statement according to the principles of the enzyme activity dependence on the temperature. In my opinion, the observed lower catalytic activity at temperatures below 30 C is definitely not related to the inhibition phenomenon.

27.  Page 11, line 461 – ‘…efficiency for the degradation’ – after that add the phrase ‘of fucoidan’.

28.  Page 12, line 476 – ‘The optimal extraction efficiency in the ultrasonication process’ change to ‘The optimal extraction efficiency for fucoidanase in the ultrasonication process’.

29.  Page 12, line 482 – ‘DEAE’ change to ‘DEAE Sepharose Fast Flow column chromatography’.

Round 2

Reviewer 1 Report

The revised manuscript is now acceptable for publication.

Reviewer 2 Report

Dear Authors

thank you for considering all my comments and suggestions. In its current form, I see no objections to further processing of the manuscript by the editorial team.